# Fe$^0$/H$_2$O Systems for Environmental Remediation: The Scientific History and Future Research Directions

**Rui Hu [1], Xuesong Cui [1], Willis Gwenzi [2], Shuanghong Wu [1] and Chicgoua Noubactep [3,*]** 

1   School of Earth Science and Engineering, Hohai University, Fo Cheng Xi Road 8, Nanjing 211100, China; rhu@hhu.edu.cn (R.H.); cuixuesong@hhu.edu.cn (X.C.); wush@hhu.edu.cn (S.W.)
2   Biosystems and Environmental Engineering Research Group, Department of Soil Science and Agricultural Engineering, Faculty of Agriculture, University of Zimbabwe, P.O. Box MP167, Mount Pleasant, Harare, Zimbabwe; wgwenzi@yahoo.co.uk
3   Department of Applied Geology, Universität Göttingen, Goldschmidtstraße 3, D-37077 Göttingen, Germany
*   Correspondence: cnoubac@gwdg.de; Tel.: +49-551-393-3191

**Abstract:** Elemental iron (Fe$^0$) has been widely used in groundwater/soil remediation, safe drinking water provision, and wastewater treatment. It is still mostly reported that a surface-mediated reductive transformation (direct reduction) is a dominant decontamination mechanism. Thus, the expressions "contaminant removal" and "contaminant reduction" are interchangeably used in the literature for reducible species (contaminants). This contribution reviews the scientific literature leading to the advent of the Fe$^0$ technology and shows clearly that reductive transformations in Fe$^0$/H$_2$O systems are mostly driven by secondary (Fe$^{II}$, H/H$_2$) and tertiary/quaternary (e.g., Fe$_3$O$_4$, green rust) reducing agents. The incidence of this original mistake on the Fe$^0$ technology and some consequences for its further development are discussed. It is shown, in particular, that characterizing the intrinsic reactivity of Fe$^0$ materials should be the main focus of future research.

**Keywords:** contaminant reduction; iron corrosion; oxide scale; water treatment; zero-valent iron

## 1. Introduction

Micrometre-size metallic iron (Fe$^0$) is one of the most commonly used materials for permeable reactive barriers (PRBs) used in groundwater remediation [1–10]. Despite its large size and low surface area, it has been successfully used for groundwater treatment with more than 200 PRBs installed worldwide [10]. However, the contaminant removal mechanisms of Fe$^0$ PRBs have not been elucidated.

Reductive degradation/precipitation using elemental iron (Fe$^0$) as a reactive medium (electron donors) to treat contaminated soils and waters has been extensively investigated during the last 28 years [11–16]. Fe$^0$ is a reducing agent (E$^0$ = −0.44 V) with reaction products (Fe$^{II}$ and Fe$^{III}$ species) which are (mostly) environmentally innocuous. Additionally, Fe$^0$ is abundantly available, for instance in the form of scrap iron and steel wool [8,10,12,17–22].

When coupled with the chemical reduction (degradation) of an oxidized contaminant (Ox), the spontaneous reduction reaction yielding a more biodegradable and/or hopefully less toxic reduced form (Red) is given by Equation (1):

$$Fe^0 + Ox \Rightarrow Fe^{2+} + Red \qquad (1)$$

Fe$^0$ permeable reactive barriers (Fe$^0$ walls) have become an established technology for in-situ groundwater remediation [10,13,14,16,23]. There is an almost consistent agreement in the literature

about direct reduction (electrons from $Fe^0$) (Equation (1)) as the main mechanism by which $Fe^0$ reduces aqueous contaminants [14,24–28]. Accordingly, Equation (1) alone is always given as the reduction scheme [28–32] and is implemented in modelling codes [13,33]. Therefore, other reductive mechanisms, including reduction by hydrogen (H/$H_2$—Equation (2)), ferrous iron ($Fe^{II}$—Equation (3)) and solid corrosion products (e.g., green rust) are considered as side effects. Moreover, contaminant adsorption onto the $Fe^0$ surface and oxide scale is acknowledged as an intermediate step towards reductive transformations [24,34–38].

$$H/H_2 + Ox \Rightarrow H^+ + Red \qquad (2)$$

$$Fe^{II} + Ox \Rightarrow Fe^{III} + Red \qquad (3)$$

However, the validity of the concept of reductive degradation/precipitation has been severely challenged in the peer-reviewed literature since 2007 [28,39–50]. A new concept was introduced, that considers adsorption and co-precipitation of contaminants within the oxide scale on $Fe^0$ as the primary removal mechanism [39–42]. The subsequent abiotic reduction is possible, but not necessarily a direct mechanism (Equation (1)). The new concept (adsorption/co-precipitation concept) is free of contradictions inherent to the reductive degradation/precipitation concept, and explains some seemingly controversial experimental results [31,32,39,40,51–54].

Noubactep [39,40] has demonstrated that the pioneers of environmental remediation using $Fe^0$ materials have not properly considered information put forth by other branches of science using $Fe^0$ materials, including; iron corrosion [55–59], organic synthetic chemistry [60,61], wastewater treatment [62,63], and the oil industry [64,65]. Furthermore, the experimental conditions used in the pioneering studies were not appropriate for traceable conclusions [39–41,66]. Despite some serious warnings [28,67–73], the inherent error of the pioneers has been perpetuated and propagated through the scientific literature [1–10,14,23,26,27,29,74,75].

The present contribution aims to demonstrate from a historical perspective that indirect reduction should have merited more attention in the $Fe^0$ remediation research than other mechanisms. The demonstration is based on knowledge available to scientists before the advent of "$Fe^0$ walls" [63,76]. The presentation is deliberately limited to the literature available before 1994 which corresponds to the first mechanistic investigations pertinent to the $Fe^0$/$H_2O$ system [24]. It is further demonstrated that; (i) iron corrosion is volumetric expansive [77] making a pure $Fe^0$ wall (100% $Fe^0$) not sustainable [78,79] due to loss of porosity and hydraulic conductivity, and (ii) the $Fe^0$/$H_2O$ system is ion-selective, making the $Fe^0$ technology more suitable for the removal of negatively charged species than neutral and positively charged ones [80–83]. Based on the preceding, the future research directions on the $Fe^0$/$H_2O$ system are highlighted.

## 2. The Interactions within $Fe^0$/$H_2O$ Systems

The voluminous literature on the aqueous $Fe^0$ reactivity under near-neutral pH conditions is characterized by a clear agreement on the formation of an oxide scale on the $Fe^0$ surface (Figure 1) [17,55,56,58–60,84–89]. There is some controversy concerning the composition and structure of this oxide scale and the mechanism by which the film breaks down [35,86,89,90]. It is well-known that the formation of the oxide scale (and its further transformation) is influenced differently by major anions ($Cl^-$, $NO_3^-$, $PO_4^{3-}$, $SO_4^{2-}$), but the mechanisms involved are unclear. In particular, there is a lack of agreement in the literature on the influence of $Cl^-$ on the oxide film. $Cl^-$ may; (i) adsorb on $Fe^0$ and avoid formation and/or adherence of oxide film, (ii) cause a thinning of the oxide film and this can lead to pitting if the potential is above a critical value, and (iii) incorporate into the film lattice and induced a subsequent local breakdown [90]. The preceding recalls that iron is corroded by water (solvent—Equation (4)) and the extent of corrosion is more or less influenced by solutes [55,58,59,64].

$$Fe^0 + 2\,H^+ \Rightarrow Fe^{2+} + H_2 \qquad (4)$$

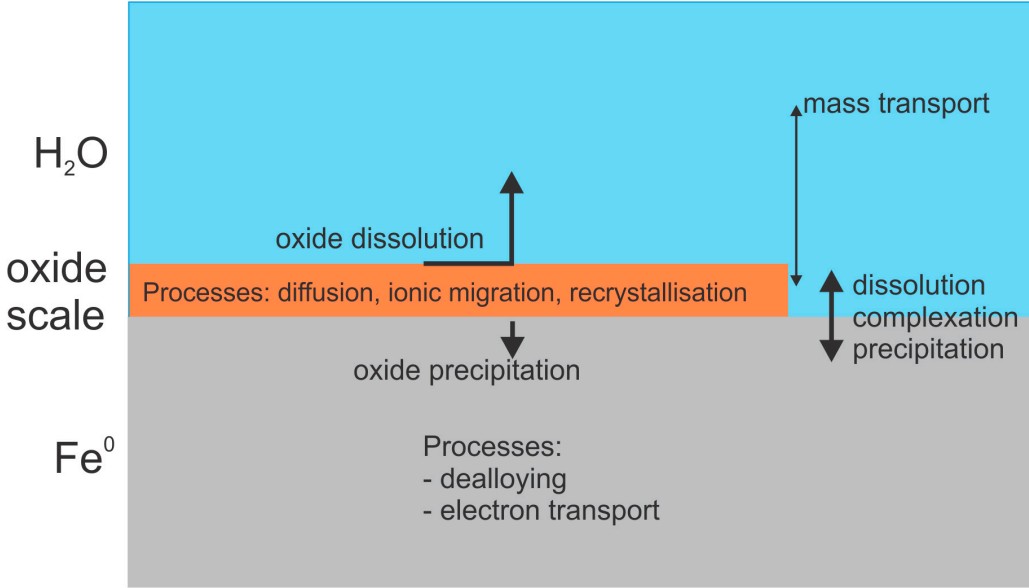

**Figure 1.** Schematic of the elemental iron ($Fe^0$)/$H_2O$ system. Features relevant to contaminant removal are labelled. Modified after Taylor [91].

Furthermore, at neutral pH values, the presence of a universal oxide scale should be considered as a rule rather than an exception [65,87]. The exception of iron oxide film breakdown should be discussed on a case-by-case basis. For example, if water contains $Cl^-$, it can be considered that the film formation is disturbed or delayed [92]. In this regard, Cohen [55] for instance, investigated the mechanism of oxide film formation on iron in aqueous solutions and the effects of oxygen ($O_2$), oxidizing ions (chromate, molybdate, nitrite, tungstate), and non-oxidizing ions (acetate, carbonate, phosphate) on this process. In the recent $Fe^0$ literature, chromate, molybdate, nitrite, tungstate, and phosphate have been investigated as contaminants in independent treatability studies, while $O_2$, acetate, carbonate, and phosphate were mostly tested as site-specific environmental variables. For this reason, one may consider Cohen's work as a seminal for an alternative and probably more accurate approach to investigate the process of contaminant removal in $Fe^0$/$H_2O$ systems. Clearly, investigating contaminant removal in $Fe^0$/$H_2O$ systems should entail characterizing the impact of individual contaminants on the process of oxide film formation (and transformation) under site-specific conditions (e.g., contaminant concentration, pH value, water velocity). This intuitive approach has not been considered, and the $Fe^0$ remediation literature is full of studies investigating the same species (including $NO_3^-$ and $PO_4^{3-}$) as contaminants in one case and co-solutes in other cases [3,9,12,16,29]. Moreover, the bulk of such studies used artificial solutions under controlled laboratory conditions (e.g., constant temperature, agitation), while those using more complex contaminated aqueous systems under real-life conditions are rare. In particular, the use of technical tools (e.g., mixing, pumping, shaking) to accelerate mass transfer at laboratory scale has generated reproducible results with little practical meaning. The major limitation of accelerated mass transfer has been the delay of the formation of the oxide scale on $Fe^0$ and, thus, minimizing the relevance of indirect reduction, while possibly favoring direct reduction at sites which would not be available under field conditions [32,44,46,67,93].

*Mass Transfer within $Fe^0$/$H_2O$ Systems*

A heterogeneous reaction involving the reduction of a contaminant in a $Fe^0$/$H_2O$ system is considered to include the following eight steps [24,91,94,95]; (i) Transport of reactants to the water/oxide interface, (ii) adsorption of reactants onto the outer oxide surface, (iii) diffusion of reactants across the oxide scale, (iv) adsorption of reactants onto the $Fe^0$ surface, (v) chemical reaction at the $Fe^0$ surface, (vi) desorption of soluble reaction products from the $Fe^0$ surface, (vii) diffusion

of reaction products across the oxide scale, and (viii) transport of soluble reaction products into the bulk solution. Figure 1 presents some of these processes. Steps (i) and (viii) are controlled by the rate of mass transfer of the soluble species (reactants and products), and hence by the hydrodynamics of the system (mixing types and intensities, water flow velocities). Step (ii) is mainly controlled by electrostatic interactions between reactants and the oxide scale. Steps (iii), (iv), (v), (vi), and (vii) are chemically controlled processes (reaction and concentration gradient) [35,63,91,93,95].

The main purpose of a kinetic study is to identify the rate-limiting or slowest step in the overall process, and how its rate may be increased. However, for brevity, kinetic aspects, and the potential interactions of individual contaminants and other dissolved species with the oxide scale (Figure 1, Table 1) are not considered herein. In essence, the whole oxide scale is a reactive system in which complexation, dissolution, precipitation, and co-precipitation occur simultaneously (Table 1), implying some parallel chemical reactions, including contaminant reduction. Barring processes occurring on bulk $Fe^0$ and in water, most of the processes occurring in the $Fe^0/H_2O$ system have not been adequately addressed in the literature (Table 1).

**Table 1.** Some relevant processes occurring in the elemental iron ($Fe^0$)/$H_2O$ system and their spatial locations. The comments relate to the extent to which each process has been considered in the $Fe^0$ literature, with "+ + +" denoting being regarded as satisfactorily, and "+" just acknowledged.

| Location | Processes | Comments |
|---|---|---|
| Bulk $Fe^0$ | dealloying, electron transport | + + + |
| $Fe^0/H_2O$ interface | Fe dissolution, complexation, precipitation | + + |
| $Fe^0$/Oxides | oxide precipitation | + |
| Oxide scale | migration of species, oxide recrystallization | + |
| Oxides/$H_2O$ | oxide dissolution/precipitation, Fe complexation | + |
| $H_2O$ | mass transfer (advection and diffusion) | + + + |

## 3. The Importance of Indirect Reduction

The primary aim of using $Fe^0$ in groundwater remediation under anoxic conditions is to exploit the negative potential of the $Fe^{II}/Fe^0$ redox couple ($E^0 = -0.44$ V) to degrade or immobilize redox amenable contaminants. However, dissolved ferrous iron from the $Fe^{III}/Fe^{II}$ redox couple ($E^0 = 0.77$ V) can act as reducing agent for some contaminants (e.g., $Cr^{VI}$) [53,62,63]. Furthermore, it has been shown that adsorbed $Fe^{II}$ (or structural $Fe^{II}$) ($E^0 = -0.34$ to $-65$ V) [96] can be more powerful in reducing contaminants than $Fe^0$ (for E < $-0.44$ V). Therefore, abiotic contaminant reduction in a $Fe^0/H_2O$ system does not necessarily take place by reduction through electrons from $Fe^0$ (direct reduction). Instead, indirect reduction (electrons from $Fe^{II}$ or even $H/H_2$) may occur and is most likely, for at least, two reasons; (i) for direct reduction to occur the contaminant should diffuse through the oxide film unless the film is conductible to cater for electron transfer, and (ii) oxide-film is a good adsorbent for both contaminant and $Fe^{2+}$ ions (resulting in more reactive structural $Fe^{II}$) [17,93]. Yet indirect reduction is currently considered as a side effect in the '$Fe^0$ remediation' literature [10,13,14]. The next paragraph demonstrates that indirect reduction (electrons from $Fe^{II}$) of nitro aromatic compounds has been known for more than 160 years and observed at neutral pH solutions since 1927.

### 3.1. Aniline and the $Fe^0/H_2O$ System

In 1854, Béchamp [97] discovered the quantitative reduction of nitro aromatic compounds to anilines by $Fe^0$ materials in acetic acid ($CH_3COOH$) [38]. The increased demand for aniline in the chemical industry led to the use of cost-effective hydrochloric acid (HCl) in place of $CH_3COOH$ (Equation (5)). Shortly afterwards, it was established that a minor fraction of the stoichiometric amount

of HCl (1/20 to 1/60) was sufficient for quantitative reduction [85]. Muspratt [98] was the first to demonstrate this experimental observation (Equations (5)–(7)):

$$C_6H_5\text{-}NO_2 + 6\ HCl + 3\ Fe^0 \Rightarrow C_6H_5\text{-}NH_2 + 3\ FeCl_2 + 2\ H_2O \tag{5}$$

$$2\ C_6H_5\text{-}NH_2 + FeCl_2 + 2\ H_2O \Rightarrow 2\ C_6H_5\text{-}NH_2, HCl + Fe(OH)_2 \tag{6}$$

$$2\ C_6H_5\text{-}NH_2, HCl + Fe^0 \Rightarrow 2\ C_6H_5\text{-}NH_2 + FeCl_2 + H_2 \tag{7}$$

As evidenced in Equations (5)–(7), $Fe^{II}$ species produced by $Fe^0$ corrosion are reducing agents for nitro aromatic compounds (indirect reduction). An innovative extension of the work of Béchamp [97] was published by Lyons and Smith [85]. These authors achieved quantitative reduction of nitro aromatic compounds by $Fe^0$ in the presence of NaCl and $FeCl_3$ (neutral salts) instead of HCl and $CH_3COOH$. In the same experiment, the efficiency of NaCl was only 84% of that of $FeCl_3$. As stated above, $Cl^-$ disturbs the formation of oxide film. Thus, the reported efficiency difference between NaCl and $FeCl_3$ primarily reflects the differential behavior of $Na^+$ and $Fe^{3+}$ in the $Fe^0/H_2O$ system and will not be further discussed here.

*3.2. $MnO_2$ and the $Fe^0/H_2O$ System*

In 2003, Noubactep and colleagues presented the reductive dissolution of $MnO_2$ as a tool to control the availability of iron corrosion products (FeCPs) in a $Fe^0/H_2O$ system [99,100]. This tool was used to establish the mechanism of aqueous U(VI) removal in the presence of $Fe^0$ [100–102]. The rationale for this application is that $Fe^0$ is not the reducing agent for $MnO_2$ reductive dissolution (Equation (8)). Contrary, $Fe^0$ oxidative dissolution (Equation (4)) is accelerated by virtue of $Fe^{2+}$ consumption by $MnO_2$ (Equation (9)) (i.e., Le Chatelier's Principle).

$$2\ Fe^0 + 3\ MnO_2 + 12\ H^+ \Rightarrow 2\ Fe^{3+} + 3\ Mn^{2+} + 6\ H_2O \tag{8}$$

$$2\ Fe^{2+} + MnO_2 + 4\ H^+ \Rightarrow 2\ Fe^{3+} + Mn^{2+} + 2\ H_2O \tag{9}$$

Equation (9) implies that FeCPs are precipitated at the surface of $MnO_2$ and not in the vicinity of $Fe^0$. This fact delays '$Fe^0$ passivation' and supports the repeatedly reported increased efficiency of $MnO_2$-amended $Fe^0/H_2O$ systems [31,50,103–107]. Results of the investigation of the reductive dissolution of $MnO_2$ using $Fe^0$ (e.g., $Fe^0/MnO_2/H_2O$ system) in hydrometallurgy corroborate the validity of Equation (9) as the main reduction path [108]. Similar to the concept of Khudenko [67], $MnO_2$ is efficient at sustaining iron corrosion and thus, the resulting parallel reactions, including indirect reduction.

Another important feature of $MnO_2$ for $Fe^0$-based filters is that its reductive dissolution is volumetric contraction [106], and $Mn^{2+}$ is more labile than $Fe^{2+}$ under subsurface conditions. This characteristic makes $MnO_2$ an excellent admixing agent to sustain the permeability of $Fe^0$-based filtration systems in the subsurface, in water treatment plants and/or in household water treatment filters.

## 4. Long-Term Permeability of $Fe^0$ Walls

Following the reductive transformation paradigm, $Fe^0$ filters have been designed based on the flux of contaminants and the stoichiometry of reductive transformations after Equation (1) (direct reduction) [9,10,13,32]. This paradigm implies that "the more $Fe^0$ is available, the more efficient the filter". This rule of thumb has not been validated [109–111]. For example, Bi et al. [110] observed that columns with less than 80% (w/w) $Fe^0$ performed better than those with higher $Fe^0$ ratios. The reason lies in the well-established knowledge that metals expand during oxidative transformation, including immersed corrosion [77–79]. In other words, the volume of each iron corrosion product (oxide) is larger than the volume of atomic iron in the metallic state ($V_{oxide} > V_{iron}$).

The proper consideration of this observation has led to the conclusion that pure $Fe^0$ filters (100% $Fe^0$) are not sustainable [78,79,112] due to loss of porosity and hydraulic conductivity over time. Proper consideration of the volumetric expansive nature of aqueous iron corrosion was possible, and even obvious in the 1990s when the first $Fe^0$ wall was constructed [10,113]. A critical review of the scientific literature at that time would have been sufficient to reveal the expansive nature of iron corrosion. In fact, gravel and sand have been routinely mixed with $Fe^0$ mostly to reduce iron costs rather than to minimize loss of porosity [110].

Independent efforts (not in the context of $Fe^0$ walls) have used steel wool to improve the efficiency of sand filters for phosphate removal [17,114–116] and suggested up to 8% $Fe^0$ to avoid clogging [115,116]. Steel wool has also been added to sand filters for better pathogen removal [117]. However, steel wool has a very low density ($g/cm^3$) [6] such that deriving the volumetric proportion from the weight proportion is not straightforward like for granular iron. These results attest that a pragmatic approach can be successful, although a science-based design is still needed.

## 5. Ion-Selective Nature of $Fe^0/H_2O$ Systems (Coulomb's Law)

Evidence from the iron corrosion literature is that the $Fe^0$ surface is permanently shielded by an oxide scale [55,58,59,65,118]. The oxide scale is made up of various iron hydroxides and oxides with positively charged surfaces at near-neutral pH values [119,120]. The fundamental Coulomb's law of electrostatics states that, a positively charged surface attracts negatively charged solutes. However, this fundamental law is silent about the redox reactivity of the solute, meaning that even the most redox reactive solutes must be attracted by the oxide scale before quantitative reductive transformation occurs. The $Fe^0$ remediation literature is currently full of examples of positively charged solutes (e.g., methylene blue) used to test the redox efficiency of $Fe^0/H_2O$ systems, another avoidable mistake [121,122].

## 6. Discussion

$Fe^0$ as a tool for remediation emerged in the early 1990s and rapidly went through the stages of laboratory testing [10,109,123], field demonstration [10,19,113,124], and commercial application [10,22,125–127]. In 2002, the US EPA designated the use of $Fe^0$ in a permeable reactive barrier (PRB) as a standard remediation technology. During this time, $Fe^0$ filtration was also established as a standard technology for municipal and/or industrial wastewater treatment [14,115,116]. However, the technology has not yet been universally accepted, partly due to the abundance of conflicting data in the literature [14,33,46,128,129]. In recent years, substantial efforts have been made to investigate such discrepancies and thereby address such knowledge gaps [14,30,31,120,124–131]. However, in some instances, such new data have added even more confusion. For example, Ruhl et al. [129] disproved the suitability of four granular inert aggregates (anthracite, gravel, pumice, and sand) to sustain the efficiency of $Fe^0$ walls. This directly contradicts the results of Bi et al. [110] discussed earlier and would mean that pure $Fe^0$ beds are sustainable. In another example, Firdous and Devlin [132] challenged the suitability of using methylene blue as a tracer to image processes occurring in dynamic $Fe^0/H_2O$ systems as presented by Miyajima and Noubactep [133]. Firdous and Devlin [132] have simply not considered the fact that Miyajima and Noubactep [133] have not used methylene blue as a model contaminant, but as a tracer to detect the availability of iron oxides in the $Fe^0/H_2O$ system.

This presentation has demonstrated that detailed processes occurring in the $Fe^0/H_2O$ system are not currently being comprehensively considered within the $Fe^0$ remediation scientific community [10,30,31,33,45,48,50]. The technology was fortuitously discovered in 1990 [134]. During the past 25 years, a myriad of tools has been applied to further constrain $Fe^0$ corrosion processes [10,14,16,135,136]. Moreover, a myriad of sophisticated analytical tools has been used to characterize experimental observations [21,127,137–140]. However, because the kinetics of iron corrosion is neither constant nor linear [141–143], a typical analytical observation (e.g., an $E_h$ measurement or XPS (X-ray photoelectron spectroscopy) analysis during or at the end of column

studies) is simply a static snap-shot, hence does not describe the processes in a holistic way. As such, it is simply impossible to generate accurate models of the dynamic processes occurring within the $Fe^0/H_2O$ system from such data, especially as these occur over an extremely variable range of time scales—from seconds to decades [142]. These difficulties are not unique to the $Fe^0/H_2O$ system [144] but are also common to other remediation technologies such as biosorption [20,28,31,46].

Available laboratory data are mostly irrelevant for field situations, mainly because of the lack of any reference $Fe^0$ material and the fact that experiments were performed under very different conditions, very far from those occurring under field conditions [31,46,71,139,142]. Additionally, it has been acknowledged that $Fe^0$ materials currently used for water treatment and environmental remediation (e.g., Gotthart-Maier GmbH or iPutech—Rheinfelden, Germany, Connelly-GPM Inc.—Chicago, IL, USA, Peerless Metals Powder Inc.—Detroit, MI, USA) have not been specially produced for these applications [10,21,145,146]. For example, Gotthart-Maier GmbH and later iPutech widely used at many test sites in Germany is a mixture of scrap materials. This means that they do not represent tailored $Fe^0$ reagents for environmental remediation. This suggests that the use of common materials (operational reference) in well-designed experiments is the first step towards improved and comparable data. It is certain, that such an approach, coupled with the consideration of the ion-selective nature of the $Fe^0/H_2O$ system [80–82] and the volumetric expansive nature of iron corrosion [77–79] will accelerate knowledge acquisition and boost the wide acceptance and application of the $Fe^0$ technology [143,147,148]. In these efforts, reduction through $Fe^{II}$ (indirect reduction) should be considered as an independent and major pathway for abiotic contaminant removal [149]. However, reductive transformation alone is not universally sufficient for contaminant removal (at $\mu g \cdot L^{-1}$ level).

Given the preceding analysis, four arguments are highlighted to advance the $Fe^0$ technology. First, adsorption onto and co-precipitation with iron oxyhydroxide and (hydr) oxide (corrosion products) enhance the removal process (Argument 1). Second, once the importance of reductive transformation for contaminant removal is accorded its rightful contribution, it is easy to accept $Fe^0$ as a generator for removing agents (oxides and hydroxides) (Argument 2). Moreover, considering that the surface of Fe oxides/hydroxides being predominantly positively charged at neutral pH values, the $Fe^0/H_2O$ system is necessarily ion-selective (Argument 3). Lastly, because iron oxides are larger in volume than the original $Fe^0$, room for expansion should be factored into the design of $Fe^0$ filters/walls (Argument 4). Arguments 1 through 4 should be considered simultaneously as they are not independent.

## 7. $Fe^0$ for Environmental Remediation

During the first 2.5 decades of $Fe^0$ technology, no concerted effort was directed at providing the appropriate $Fe^0$ material for utility engineers [16,128,130,136,139]. Despite a remarkable success [10,15,21,22,116], the $Fe^0$ research community should continue to seek better and new ways to make an already proven efficient and affordable technology more reliable [10,15,45,140].

The most important area of improved reliability is arguably the field of material selection. Characterizing the intrinsic reactivity of used materials should increasingly come to the forefront. Factors influencing the rate of iron corrosion and, thus, the remediation efficiency of $Fe^0/H_2O$ systems including $Fe^0$ type, water chemistry (e.g., local hydrogeology or admixing agents), and operation conditions (e.g., $Fe^0$ mass, volumetric ratio of $Fe^0$ in the mixture, temperature, thickness of the reactive layer ($Fe^0$-based), water flow rate/hydraulic retention time) should be considered. Material selection is further complicated by the diversity in the production quality of available $Fe^0$ types and the lack of linear correlation between the intrinsic reactivity and the elemental composition of $Fe^0$ materials [21,138–140,150–152].

Available knowledge on aqueous corrosion science is wide and includes cementation, organic synthesis and results reported in utility pipes from the oil industry and water/wastewater reticulation systems [14,33,46,63]. For example, cementation studies and Béchamps-like reactions have been carried out for more than 150 years. Their fundamental understanding can be adapted to the $Fe^0$ remediation industry, mainly characterized by lower concentration solutions and higher pH values.

In particular, it is not acceptable, that 30 years of research on $Fe^0$ remediation has not characterized used materials. In essence, the research performed by $Fe^0$ remediation researchers is comparable to that used to develop hydrometallurgical methods for precipitating a target metal from a given source as economically as possible (on a case-to-case basis) [23,147,153,154]. However, a caveat to this approach is that this knowledge is not directly transferable to the $Fe^0$ remediation industry because the size (e.g., diameter) of $Fe^0$ materials used in environmental remediation is typically smaller than the thickness of the pipes used in the oil industry or for water pipes (Table 2) [142]. To ensure the reliability of $Fe^0$ walls and household water filters, targeted replacement for depleted (corroded) materials is necessary. For a risk-based maintenance or replacement strategy of $Fe^0$ walls in service, it is imperative to relate the $Fe^0$ intrinsic reactivity to hydro-geochemical conditions as a standard practice for a geo-environmental engineer. Such standard practice or guidelines should be formulated for specific applications, e.g., 'use a $Fe^0$ with an average corrosion rate of xy for carbonate-rich waters' or 'use a $Fe^0$ with an average corrosion rate of yz for chloride rich waters'. Based on such guidelines, process-based models for the design of appropriate materials can be developed and used by researchers, engineers and asset managers. Such tools for asset management of $Fe^0$ walls would introduce a systematic approach in the design of new $Fe^0/H_2O$ systems and the evaluation of those already in operation.

**Table 2.** Chronology of selected historical observations showing that the science of the $Fe^0/H_2O$ system was explored before the 1970s while cementation studies and Béchamps-like reactions have been carried out for more than 150 years.

| Year | Event |
|---|---|
| <1850 | The cementation reaction is used for winning metals from ores [155–157] |
| <1850 | Iron shavings are used to treat drinking water [147,158,159] |
| 1854 | Béchamp synthesized of aniline from nitrobenzene and $Fe^0$ (iron and organic acid) [97] |
| 1865 | Bekelov suggested that all cementation reactions are electrochemical in nature [155,157] |
| 1873 | Bischoff established the spongy iron filter for household [147,159,160] |
| 1881 | Spongy iron filters are tested at large scale in Antwerp (Belgium) [147,158,160] |
| 1883 | Spongy iron filters secured water supply in Antwerp (Belgium) [147,160–162] |
| 1885 | Revolving purifiers are installed in Antwerp (Belgium) [147,161,162] |
| 1888 | Muspratt rationalized the successful use of HCl in the Béchamp reduction [98] |
| 1914 | Holt used scrap iron instead of coarse scrap iron for the cementation of $Pb^{II}$ [98,155,157] |
| 1923 | Lueg showed that aniline and other substances inhibit iron corrosion [98,155,157] |
| 1928 | Oldright and co-workers showed that only thin $Fe^0$ beds are long-term sustainable [155] |
| 1928 | Knowlton reported that the rate of iron corrosion is higher in NaCl solutions [60] |
| 1928 | Knowlton reported that the used $Fe^0$ type determines the extent of reduction [60] |
| 1951 | Lauderdale and Emmons used steel wool to remove radioactive species from water [153] |
| 1951–1961 | Werner published almost yearly review articles on "Amination by Reduction" [163–169] |
| 1951–1961 | The Béchamp reduction is extended to other groups of compounds [163–169] |
| 1969 | Case and Jones treated $Cr^{VI}$- and $Cu^{II}$-containing brass mill effluents with scrap iron [170] |
| 1984 | Tseng et al. used steel wool to concentrate $^{60}Co$ from nuclear effluent [154] |
| 1986 | Harza Environmental Services patented Se(VI) removal in $Fe^0$ beds [171] |
| 1991 | Khudenko established the cementation based reductive degradation of organics [67] |
| 1990 | Reynolds and co-authors observed dechlorination of RCl in $Fe^0$-based vessels [11] |
| 1994 | $Fe^0$ is established as an efficient material for subsurface reactive walls [1,10,113] |

## 8. Lessons from the History and Future Directions

There are two important features from Table 2 for the future of $Fe^0$ filters for water treatment and safe drinking water provision in particular: (i) The technology is old and has been successfully used both at household and large scale for water treatment [147,151,171–176], and (ii) the mode of operation of $Fe^0/H_2O$ system was established in 1991 [67]. The ancient use of $Fe^0$-based systems for safe drinking water provision is summarized by Mwakabona et al. [147] and will not be discussed herein. It is just recalled that these systems were very efficient to remove pathogens and organics (humic substances). Accordingly, properly designed $Fe^0/H_2O$ systems will efficiently treat water as

research drawn from the last two to three decades has demonstrated their suitability for all classes of chemical contamination [10,12,16].

In May 1991, Boris Mikhail Khudenko published a concept paper in 'Water Science and Technology' entitled 'Feasibility evaluation of a novel method for destruction of organics'. Khudenko [67] where, the author presented cementation as a tool to accelerate $Fe^0$ oxidative dissolution (Equation (4)) and produce H species for the reductive transformation of organic species. In other words, the oxidation–reduction of organics is induced as a parallel reaction to this cementation reaction. A SCOPUS search on 19 October 2018 indicated that Khudenko [67] has been only referenced eight (8) times (Table 3) [177–181] and has not been considered in some literature discussing the removal mechanisms of organics in $Fe^0/H_2O$ systems [10,24,25,182]. The concept of Khudenko [67] was based on a profound understanding of the $Fe^0/H_2O$ system [183–185] and corresponds to Noubactep's concept [39,40] that has been difficult to accept [48,49]. It is hoped that this independent proof will end the mechanistic discussion and orient all energies to design the next generation of efficient $Fe^0/H_2O$ systems for water treatment, including $Fe^0$ filters.

To further validate the concepts presented here and improve the mechanistic understanding of $Fe^0/H_2O$ remediation systems further research should address the following:

(1) Understanding the role and mechanisms of interfering inorganic and organic species typically occurring in natural multi-component aqueous systems under relevant environmental conditions,

(2) Elucidating the processes occurring on the various material phases (solid, liquid, and solid–liquid interface), and their effects on the formation and persistence of the iron oxide film. Recent advances in surface analytical techniques for solid-state characterization enable such detailed studies,

(3) Long-term studies using typical multi-component contaminated aqueous media conducted in a quiescent mode are required to overcome some of the limitations associated with short-term studies based on artificial solutions and ideal experimental conditions (e.g., agitation, constant temperature),

(4) Development and evaluation of tailor-made $Fe^0/H_2O$ systems that accounts for the expansive nature of iron corrosion, ion-selectivity, and the role of co-solutes/agents. This will overcome the limitations associated with the use of materials not purposively developed for $Fe^0$ remediation.

**Table 3.** SCOPUS chronological bibliometry of Khudenko [67] showing citation by eight (8) sources comprising four (4) review and four (4) research articles.

| Year | Journal | Type | Section | Reference |
|------|---------|------|---------|-----------|
| 2015 | Water Research | Review | Introduction | [16] |
| 2014 | Environmental Science & Technology Letters | Research | Discussion | [75] |
| 2010 | ACS Symposium Series | Review | Introduction | [177] |
| 2009 | Chemosphere | Research | Discussion | [178] |
| 2007 | Environmental Science & Technology | Research | Introduction | [179] |
| 2004 | Environmental Science & Technology | Research | Discussion | [145] |
| 2000 | Environmental Technology | Review | Introduction/discussion | [180] |
| 1998 | ACS Symposium Series | Review | Discussion | [181] |

## 9. Conclusions

The real challenge for active researchers on $Fe^0$ for water treatment and environmental remediation is to properly use knowledge available in the mainstream corrosion science (Table 2) to design efficient $Fe^0/H_2O$ systems. This task can be supported by available modern analytical tools (including surface-analytical techniques) to investigate well-known processes under relevant environmental conditions. However, the key lies in considering the system as a dynamic one comprising adsorbing, complexing, oxidizing and reducing species working in synergy for water decontamination (Arguments 1 through 4, Section 3). The three decades old work of Khudenko [67]

recalls that scientific progress is at best achieved when the state-of-the-art knowledge is accurately given. Alternatively, a science-based approach can enable the rediscovery of the wheel. The fact that, one decade ago Noubactep [39] had independently achieved the conclusions of Khudenko [67] is regarded as the last proof for any researcher who might be impressed by the large volume of publications considering $Fe^0$ as a reducing agent under environmental conditions. Despite three (1991–2018) or one (2007–2018) lost decade(s), future work should focus on directly investigating the highlighted arguments (1 to 4) to uncover the potential of the $Fe^0$ technology and specify its limitations on a scientific basis. To this end, specific thematic areas warranting further research were highlighted.

**Author Contributions:** R.H., X.C., W.G., S.W. and C.N. contributed equally to manuscript compilation and revisions.

**Funding:** The work is supported by the Ministry of Education of the People's Republic of China through the Program "Research on Mechanism of Groundwater Exploitation and Seawater Intrusion in Coastal Areas" (Project Code 20165037412) and by "the Fundamental Research Funds for the Central Universities" ("Research on the hydraulic tomographical method for aquifer characterization", Project Code 2015B29314). It is also supported by Jiangsu Provincial Department of Education, Project Code 2016B1203503.

**Acknowledgments:** The author thanks Richard A. Crane (University of Cardiff, Cardiff, UK) and Mohammad A. Rahman (University of Hannover, Hannover, Germany) for helpful discussions, review of the manuscript, and the illustrative material. We acknowledge support by the German Research Foundation and the Open Access Publication Funds of the Göttingen University.

**Conflicts of Interest:** The authors declare no conflict of interest.

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
