# Peer review of "Fe0/H2O Systems for Environmental Remediation: The Scientific History and Future Research Directions"

_water, doi:10.3390/w10121739_

Reviewer 1 Report

The manuscript entitled "Fe0/H2O systems for environmental remediation: The scientific history and future research directions" in my opinion, is of particular interest review paper. 

The authors state with precision and clarity the review's objectives and goals and argue with a very critical view.

Furthermore the authors have made proper literature review using the appropriate references, highlighting both the key scientific findings and deficiencies exist.

Particularly important for this review paper is that it results in concrete proposals for future work.

Ιn my view it is a complete study and I suggest that this manuscript can be accepted for publication in its present form.

Author Response

Many thanks for this evaluation.

Reviewer 2 Report

The review articles provide good overview on the state of the art of Fe0 technology for remediation. The article discussed the existing conflict in the underlying mechanisms of the decontamination (direct versus in indirect) as well as the future outlook of the technology. Hence, the article is suitable to be published in the journal.

Nevertheless, major revision in the structure of the review paper is required. The author didn’t outline the problem upfront (e.g. consider moving the first paragraph in Discussion to the beginning of the article). I don’t think the sub-headings are informative e.g. There is no point of making subsection 2.1 or 3.1 if 1 if there only one section and there wasn’t any 2.2. or 3.2.

The structure of the review paper usually doesn’t have a discussion part. So, it is better to compress the discussion part with some of section in the introduction. As I felt that the structure is a bit scattered, e.g. collapse the support for discrepancy in mechanisms in one section.

Better to provide the history/timeline as Figure instead of Table.

The future outlook can be combined with the conclusion at the end.

Author Response

English language and style: (x) Moderate English changes required, Done, thanks!

Comments and Suggestions for Authors:

The review article provides good overview on the state of the art of Fe0technology for remediation. The article discussed the existing conflict in the underlying mechanisms of the decontamination (direct versus in indirect) as well as the future outlook of the technology. Hence, the article is suitable to be published in the journal.

Many thanks for this evaluation.

Nevertheless, major revision in the structure of the review paper is required.

Comments 1: The author didn’t outline the problem upfront (e.g. consider moving the first paragraph in Discussion to the beginning of the article).

We have now added a short introductory paragraph, but have not changed the Discussion. The Discussion was conceived to answer the question 'So what?'

Comments 2: I don’t think the sub-headings are informative e.g. There is no point of making subsection 2.1 or 3.1 if 1 if there only one section and there wasn’t any 2.2. or 3.2.

Thanks for this remarks! It was 3.1 (under 3. The Importance of Indirect Reduction). We have added '3.2. MnO2and the Fe0/H2O system' as another proof that parallel reactions are occuring.

Comments 3: The structure of the review paper usually doesn’t have a discussion part. So, it is better to compress the discussion part with some of section in the introduction. As I felt that the structure is a bit scattered, e.g. collapse the support for discrepancy in mechanisms in one section.

We fully understand this view. Hovewer the manuscript was prepared to answer many open questionsand structured accordingly. For example it was important to take the reader on the popular path before showing the alternative. As stated above, Discussion is just and answer to 'So What?'. It also seeks to present a synthesis of the various aspects highlighted.

Comments 4: Better to provide the history/timeline as Figure instead of Table.

The data are too large for a readible graphical timeline.

The future outlook can be combined with the conclusion at the end.

Thanks for this suggestion. We considered the suggestion. However, it is our opinon that, since ‘Future research directions‘ is included in the title, we need a section on that so that it stands out. Moreover, including the future research in the Conclusion will make it very long. Therefore we made no corrections.